# Osteogenic-Inducing Apatite/Agarose/Gelatin Hybrid Scaffolds Embedding Gold Nanoparticles

**DOI:** 10.3390/pharmaceutics17091103

**Published:** 2025-08-25

**Authors:** María Victoria Cabañas, Paola S. Padilla, Mónica Cicuéndez, Sandra Sánchez-Salcedo, Jesús Román, Juan Peña

**Affiliations:** 1Departamento de Química en Ciencias Farmacéuticas, Facultad de Farmacia, Universidad Complutense de Madrid, UCM, 28040 Madrid, Spain; vcabanas@ucm.es (M.V.C.); paolapadillava@uide.edu.ec (P.S.P.); mcicuend@ucm.es (M.C.); sansanch@ucm.es (S.S.-S.); 2Instituto de Investigación Hospital 12 de Octubre, i+12, 28040 Madrid, Spain; 3Instituto de Investigación Sanitaria del Hospital Clínico San Carlos (IdISSC), 28040 Madrid, Spain; 4CIBER de Bioingeniería, Biomateriales y Nanomedicina (CIBER-BBN), 28040 Madrid, Spain

**Keywords:** functionalized (PEGlyated) gold nanoparticles, bone tissue engineering scaffolds, hydrogels, mineralization, circularity index

## Abstract

**Objectives**: To prepare porous scaffolds combining hydrogel and hydroxycarbonateapatite, enriched with a promising therapeutic agent, gold nanoparticles, to improve bone regeneration. The fabrication procedure is conducted under mild conditions, without toxic or aggressive chemicals, at physiological pH, and low temperatures; **Methods**: Gold nanoparticles (15–20 nm), were obtained by the Turkevith method. The scaffolds were fabricated by the GELPOR3D method, which has demonstrated its ability to integrate thermal labile molecules, during the scaffold fabrication process. The role of these nanoparticles in promoting cell adhesion, proliferation, and mineralization processes in vitro has been studied using osteoprogenitor MC3T3-E1 cells; **Results**: The scaffold fabrication conditions, combined with the surface functionalization of the gold nanoparticles with poly(ethylene glycol), ensure their uniform distribution throughout the scaffold and facilitate their gradual release over 48 h in a physiological medium. A significant increase in the mean cell area and a significant decrease in the circularity index during the early stages of osteoblast differentiation are observed. These pieces of evidence suggest that adequate cell spreading could lead to enhanced proliferation and matrix deposition activity; **Conclusions**: Scaffolds containing these gold nanoparticles exhibited a marked improvement in adhesion, proliferation, and mineralization of preosteoblasts (MC3T3 cells) at the concentrations studied. The functionalization of the nanoparticles, along with the shaping procedure employed, is critical for their homogeneous dispersion throughout the scaffold and their progressive release. The findings confirm the crucial role of gold nanoparticles in the early stages of osteoblast differentiation, which is essential for the transition from premature osteoblasts to mature osteoblasts.

## 1. Introduction

The use of porous scaffolds as three-dimensional networks for cell adhesion is a promising strategy for bone tissue regeneration. Ideal scaffolds for bone tissue engineering should be biocompatible, biodegradable, and possess a porous architecture with mechanical properties similar to those of natural bone. Depending on their composition, these scaffolds can also stimulate osseoinduction and osseointegration [1,2,3]. However, no single material meets all these requirements, leading researchers to explore composite scaffolds that leverage the strengths of various materials. Scaffolds that combine biopolymers and bioactive ceramics are considered particularly promising alternatives [4,5,6]. Additionally, incorporating a drug or bioactive molecule whose controlled release facilitates the scaffold’s integration and performance has become a milestone in the design and fabrication of this type of construct.

Hydrogels, especially those based on natural polysaccharides, such as agarose, have garnered attention due to their resemblance to the extracellular matrix, excellent biocompatibility, and thermosensitive properties [7,8,9,10]. Furthermore, the incorporation of another type of hydrogel, gelatin—a type I collagen derivative—enhances the bioactivity of agarose-based scaffolds by introducing cell recognition motifs that facilitate tissue integration [11,12]. Adding calcium phosphates, such as hydroxyapatite, further improves the mechanical performance and bioactivity of hydrogel-based scaffolds, promoting osteoinduction and osteoconduction. Notably, carbonate-substituted hydroxyapatite displays superior bioactive properties compared to stoichiometric hydroxyapatite [13,14,15].

Scaffold architectures that mimic the hierarchical, interconnected porous structure of bone are essential for bone cell development. Techniques, such as the GELPOR 3D method [16,17], enable the fabrication of scaffolds with customized porosity and the incorporation of thermally sensitive substances, including proteins, nanoparticles, and drugs. This method’s advantages—simplicity and low cost—are complemented by the ability to incorporate these substances under physiological conditions, such as low temperatures and neutral pH, without toxic or aggressive solvents. This simultaneous incorporation ensures a homogeneous distribution and integration, resulting in a more controlled and progressive release [18,19,20]. These scaffolds demonstrate excellent surgical handling, mechanical performance, integration, and gradual degradation when implanted subcutaneously in rats [21], showing no premature decomposition that could compromise their performance [12,22].

Bone morphogenetic protein-immobilized scaffolds have been widely used in bone tissue engineering [23,24], but their clinical application has been limited due to challenges such as low yields, high costs, unwanted ectopic bone formation, and local inflammatory reactions. These facts have prompted the search for alternative strategies that offer improved cost-effectiveness and functional stability for osteoinductive agent-immobilized bone substrates.

An alternative strategy is to incorporate nanomaterials, such as liposomes, metal, ceramic, or polymeric nanoparticles, into the scaffolds [25,26,27,28]. In this regard, gold nanoparticles (GNPs) have been widely studied for drug release or tissue engineering, more specifically for bone regeneration [29,30,31,32,33,34,35]. Several studies have reported that GNPs can stimulate various biochemical signaling pathways, including p38 MAPK, autophagy, and the Wnt/β-Catenin Pathway [36,37]. These nanoparticles enhance osteogenic gene expression, alkaline phosphatase activity, and mineralized nodule formation, with both in vitro and in vivo studies demonstrating rapid and effective bone regeneration [35,38,39].

In this study, we present a novel composite scaffold that combines agarose, gelatin, and carbonate-substituted hydroxyapatite, enriched with gold nanoparticles. Our goal was to develop cost-effective, stable, and osteoinductive 3D scaffolds that do not utilize toxic chemicals or solvents. We evaluated the release of gold nanoparticles under physiological conditions and their role in promoting cell adhesion, proliferation, and mineralization processes in vitro, highlighting their potential to advance bone tissue engineering.

## 2. Materials and Methods

### 2.1. Synthesis and Characterization of Gold Nanoparticles

Gold nanoparticles (GNPs) were synthesized by reducing tetrachloroauric(III) acid trihydrate with a trisodium citrate dihydrate solution using the method described by Turkevith [40,41]. Briefly, a 50 mL solution of 0.5 mM HAuCl_4_ (Sigma–Aldrich, Steinheim, Germany) was prepared and brought to a boil with vigorous stirring. Once the temperature stabilized, 1 mL of a 2% aqueous trisodium citrate solution (Sigma–Aldrich, Steinheim, Germany) was added. The mixture was stirred for 15 min until the solution exhibited a ruby-red coloration. After the color change, the solution was removed from the heat source and allowed to cool while stirring for 24 h. The resulting GNPs suspension was then centrifuged at 14,000 rpm at 4 °C, washed twice with distilled water, suspended in Phosphate Buffered Solution (PBS), and stored at 4 °C for later use.

PEGylated GNPs (GNPs-PEG) were created by conjugating the GNPs with poly(ethylene glycol) methyl ether thiol, PEG-SH (Ernst-Simon-Strasse, Tuebingen, Germany, Mw = 1218.1037). This functionalization was conducted at room temperature by adding 5 mL of PEG-SH solution (7.5 mg/mL) to 10 mL of the GNPs suspension and stirring for 30 min. The resulting GNPs-PEG suspension was centrifuged at 14,000 rpm at 4 °C, washed twice with distilled water, resuspended in 5 mL of PBS, and stored at 4 °C for later use.

The suspensions of both GNPs and functionalized GNPs-PEG were characterized using UV–visible spectrophotometry (UV–vis) in the range of 400–800 nm with a SynergyTM 4 plate reader (BioTek Instrument, Winooski, VT, USA). Their morphology and size were analyzed using Transmission Electron Microscopy (TEM) on a JEOL 1200EX. Hydrodynamic size (DLS) and zeta potential were measured in distilled water using a Malvern Zetasizer Nano ZS90 (Malvern Instruments, Malvern, UK). The concentration of gold nanoparticles in the suspension was determined through inductively coupled plasma-optical emission spectrometry (ICP-OES) using a PerkinElmer Optima 2100 DV ICP apparatus (Waltham, MA, USA). Prior to the analysis, samples were digested overnight at 90 °C in a mixture of HNO_3_ and HCl (1:3) and subsequently diluted with distilled water.

### 2.2. Scaffolds Manufacturing and Characterization

The scaffolds were fabricated using the GELPOR3D method [16,22] as described in Figure 1. For this purpose, agarose powder (Sigma–Aldrich, Steinheim, Germany, for routine use) was suspended in PBS solution (3.5% *w*/*v*) at pH 7.4 and heated to 90 °C with continuous stirring until a translucent suspension was achieved. The temperature was then reduced to 40 °C before adding gelatin type A (Sigma-Aldrich), ceramic hydroxycarbonateapatite (HCA, synthesized in our laboratory [42]), and the gold nanoparticle suspension. Once the slurry was homogeneous, it was poured into a mold and allowed to gel at room temperature for approximately 5 min. Subsequently, the filaments were withdrawn and the resulting 3D interconnected porous scaffold removed [43] (Figure 1).

The scaffold composition was based on previous results and consisted of (% by weight): 45% agarose, 10% gelatin, and 45% hydroxycarbonateapatite. Two different volumes (1 or 4 mL) of an aqueous gold NP suspension (7.8 μg/mL) were incorporated into the agarose mixture using a syringe: AGH-1Au and AGH-4Au. Additionally, a sample without gold nanoparticles (AGH) was prepared. Scaffolds were also produced without the ceramic component to aid in characterizing the inclusion of gold nanoparticles (AG, AG-1Au, and AG-4Au).

The obtained scaffolds are hydrogels with a high water content (>90%, as determined by Thermogravimetric Analysis) and were used fresh for the various tests conducted. For structural characterization, these scaffolds were freeze-dried and analyzed using Scanning Electron Microscopy (SEM) on a JEOL JSM7600F equipped with an Energy-Dispersive X-ray Analysis (EDS) detector, X-Ray Diffraction (XRD) on a Philips X’Pert Plus, Cu Kα diffractometer, and Fourier Transform Infrared Spectroscopy (FTIR) on a Thermo Nicolet Nexus 470 spectrometer.

### 2.3. In Vitro Studies

#### 2.3.1. Gold Nanoparticle Release Assays

The release study of GNPs-PEG from the scaffolds was conducted under physiological conditions (pH = 7.4 and 37 °C) using a PBS solution. In each of the triplicate experiments, a scaffold (12 × 2 mm) was immersed in 2 mL of PBS in 12-well plates under mechanical oscillation. The concentration of released gold NPs over time was determined by UV–vis spectrophotometry, measuring the absorbance of the supernatant at λ = 520 nm. A calibration curve ranging from 0.1 to 4 μg/mL was created beforehand to determine the concentration of gold nanoparticles.

#### 2.3.2. Cell Assays

##### Culture of MC3T3-E1 Preosteoblasts

MC3T3-E1 cells were cultured in alpha-MEM (Gibco, Thermo Fisher Scientific, Wilmington, DE, USA) supplemented with 10% fetal bovine serum (FBS) (Gibco), 1 mM L-glutamine (Gibco), and 1% penicillin/streptomycin (Gibco) at 37 °C under a 5% CO_2_ atmosphere. Cells were routinely subcultured every 2–3 days, when they reached 80% confluency, using a trypsinization method.

##### Cell Morphology and Adhesion Assay

Before performing the cell adhesion assay, scaffolds were sterilized under ultraviolet radiation for 1 h. MC3T3-E1 cells were then seeded onto the surface of the hydrogels at a density of 2 × 10^4^ cells per scaffold (in 12-well plates) and incubated for 9 days in culture medium without any specific differentiation factors. Phalloidin-rhodamine conjugate (red) was used to stain the F-actin filaments of the cytoskeleton, while DAPI (4′,6-diamidino-2-phenylindole, a blue fluorescent marker; Fluoroshield (Sigma-Aldrich, St. Louis, MO, USA) was used to stain the cell nuclei. After removing the culture medium and washing the cells twice with PBS, 0.5 mL of a 4% paraformaldehyde (PFA) solution (Sigma-Aldrich) was added per well to fix the cells, followed by incubation in an oven at 37 °C for 20 min. The PFA was then thoroughly washed out with PBS, and 1 mL of PBS was left in each well. After this, 20 μL of phthalocyanine was added and incubated for 20 min with agitation in darkness. DAPI staining was then performed at a dilution of 1:5000 for 10 min. Finally, samples were observed using the Evos FL Cell Imaging System (Advanced Microscopy Group, Bothell, WA, USA). Control scaffolds (AGH without GNPs-PEG) were also included.

For fluorescence microscopy, an EVOS FL Cell Imaging System inverted optical microscope equipped with three types of LED light (IEX (nm); IEM (nm)) from AMG (Advanced Microscopy Group, Bothell, WA, USA) was used. The red channel was used to observe the cytoskeleton labelled with the probe Phalloidin-rhodamine conjugate (540/565) and the blue channel to observe the cell nucleus labelled with DAPI (358; 461).

For quantitative analysis of cell morphology, images were captured from triplicate samples (10 randomly chosen fields of view each) on a given number (n = 50) of randomly chosen cells. Cell circularity is used to provide a quantification of cell shape. The formula used for determining the circularity index (CI) is CI = 4π*(Area/Perimeter^2^). A value of 1.0 indicates a perfect circle and a value of 0.0 a totally elongated structure.

##### Cell Proliferation Assay

Cells were seeded onto the surface of the hydrogels at a density of 2 × 10^4^ cells per hydrogel (in 12-well plates) and incubated for 1, 6, and 8 days in culture medium without any specific differentiation factors. Cell proliferation was assessed using the Alamar Blue test (Invitrogen™ Thermo Fisher Scientific, Waltham, MA, USA), with fluorescence measured at 560–570 nm using the Synergy™ 4 reader (Biotek). Control scaffolds (AGH without GNPs-PEG) were included.

##### Mineralization Assay

Mineralization was assessed using alizarin red S (ARS) staining, indicated by the formation of calcium nodules in MC3T3-E1 cells. After washing the wells with PBS, the cells were fixed with 70% ethanol for 1 h at 4 °C. Subsequently, the cells were treated with a 40 mM ARS solution (0.136 g/10 mL) at pH 4.2 for 30 min, followed by washing with PBS to remove non-specific staining. To quantify calcium mineralization, the dye was eluted using 10% (*w*/*v*) cetylpyridinium chloride (Sigma-Aldrich) in 10 mM sodium phosphate buffer at pH 7 for 30 min, and the absorbance was measured at 620 nm using the SynergyTM 4 reader (Biotek Instrument). Control scaffolds (AGH without GNPs-PEG) were also included.

##### Statistical Analysis

Data are expressed as means ± standard deviations from three independent experiments, each carried out in triplicate. Statistical analysis was performed using the Statistical Package for the Social Sciences (SPSS) software, version 22. Comparisons among groups were made using analysis of variance (ANOVA), with the Scheffé test employed for post-hoc evaluations. In all the statistical evaluations, a *p*-value of <0.05 was considered statistically significant.

## 3. Results and Discussion

### 3.1. Gold Nanoparticles

The ruby-red color of the gold nanoparticle (GNP) suspension (Figure 2) provides critical qualitative information. A purple coloration indicates the aggregation of the gold nanoparticles. The UV–vis absorption spectrum of these red GNPs (Figure 2) displays a localized surface plasmon resonance (LSPR) at 520 nm, which is characteristic of spherical GNPs with sizes ranging from 15 to 20 nm [44,45]. Additionally, the sharp absorption band suggests that these GNPs are uniformly distributed and not aggregated.

To determine the size and shape of the GNPs by TEM, a drop of an aqueous suspension containing the nanoparticles was placed on a copper grid and allowed to evaporate. A TEM image of the GNPs is shown in Figure 3a, revealing a quasi-spherical morphology with a monomodal size distribution. The statistical size was measured to be 14 ± 1.9 nm using ImageJ software version 1.54k, based on measurements of 130 nanoparticles. Additionally, the inset in the figure shows that the average hydrodynamic size obtained from DLS was 15.60 ± 0.3 nm, indicating satisfactory dispersibility [46]. The zeta potential of the prepared GNPs was found to be −15.6 mV, which can be attributed to the negative charge of the citrate groups on the GNPs’ surface.

These freshly prepared GNPs exhibit stability; however, aggregation occurs after four days. To enhance their stability, these nanoparticles were functionalized with PEG-SH. The UV–vis spectrum of the modified nanoparticles (GNPs-PEG) also exhibits LSPR at 520 nm, maintaining the spherical shape and size, as indicated in the TEM image in Figure 3b. The GNPs-PEG possess a zeta potential of −20.5 mV, similar to the non-functionalized gold nanoparticles, but with a higher hydrodynamic size of 24.36 ± 3.1 nm due to the conjugation of PEG-SH groups onto the GNPs’ surface (Figure 3b).

### 3.2. Scaffolds Manufacturing and Characterization

Agarose, a thermogelling polysaccharide, can form a gel at room temperature within minutes without requiring chemical crosslinking agents. This biocompatible polysaccharide is ideal for fabricating scaffolds under physiological conditions. However, since agarose lacks cell-recognition motifs, the bioactivity of such composites must be significantly enhanced by incorporating specific proteins. In fact, we recently demonstrated that the addition of gelatin to apatite/agarose scaffolds improves their interaction with New Zealand rabbit MSCs [12].

Despite the findings by Tentor et al. [47], which indicated that the introduction of GNPs into chitosan/pectin hydrogels affects the gelation temperature, our study found that the incorporation of gold nanoparticles in the amounts analyzed did not affect the agarose gelation process, which is crucial for scaffold fabrication. The presence of these nanoparticles within the scaffold was confirmed by the color change from white to ruby-red (Figure 1b). The homogeneous distribution of the nanoparticles within the scaffold was apparent to the naked eye due to the ruby-red coloration previously observed during the preparation of the nanoparticles. It was essential to cool the agarose suspension to 40 °C; otherwise, gold particle aggregation was observed, evidenced by the color change of the scaffold from ruby-red to purple, which had also been noted during the optimization process of gold nanoparticles.

Purple agarose scaffolds were observed when non-functionalized GNPs were used, indicating that these gold NPs tend to aggregate within the scaffold matrix (Figure 4a). These aggregated gold particles could not be released from the scaffold when immersed in a PBS solution, as shown by the absence of the characteristic LSPR in the UV–vis absorption spectrum of the supernatant collected after 12 h of immersion (Figure 4a).

In contrast, when GNPs-PEG were employed for scaffold fabrication, the aggregation problem was eliminated, resulting in ruby-red-colored scaffolds (Figure 4b). Furthermore, these scaffolds released the functionalized gold nanoparticles upon contact with an aqueous medium, as confirmed by the appearance of an absorption maximum at 520 nm in the UV–vis spectrum of the supernatant (Figure 4b).

Therefore, the functionalization of GNPs with PEG-SH promotes proper and stable incorporation into the scaffolds, allowing for nanoparticle release in a physiological medium. This phenomenon highlights the significance of color variation as a marker of undesirable gold particle aggregation.

The XRD patterns of scaffolds containing gold NPs are similar to those of scaffolds without gold nanoparticles. They exhibit diffraction maxima corresponding to carbonate apatite and an amorphous structure associated with agarose and gelatin [43]. Additionally, as the scaffolds were prepared in PBS, maxima corresponding to NaCl were also detected (see Appendix A). The presence of GNPs-PEG cannot be identified using FTIR spectroscopy, as only bands attributed to agarose, gelatin, or hydroxycarbonateapatite (HCA) are observable (see Appendix A).

Regarding the characterization of the scaffold morphology, Figure 5 presents photographs of scaffolds fabricated with and without ceramic components. The scaffolds exhibit a homogeneous coloration that intensifies with increasing GNPs-PEG concentration, suggesting an even distribution of nanoparticles within the hybrid matrix. This figure also includes magnified images of the scaffold interior, revealed by cutting the surface, which allows for the quantification of the 900 µm channels, their orientation in three spatial dimensions, and their connectivity.

Scanning Electron Microscopy (SEM) further enhances the morphological characterization of the scaffolds at much higher magnifications. Figure 6 shows the complete integration of ceramic particles within the agarose/gelatin matrix. Specifically, Figure 6a depicts the characteristic honeycomb morphology of the lyophilized gels, resulting from the sublimation of ice crystals during the freeze-drying process. As discussed in our previous publications [19,43], lyophilization not only serves as a preservation method but also creates additional porosity, resulting in 100–200 µm parallel pores that contribute to better scaffold integration and performance. Figure 6b confirms that the ceramic particles are fully embedded within the agarose/gelatin matrix, which forms the walls of the honeycomb cells. Backscattered electron SEM images at varying magnifications (Figure 6c,d) reveal the presence of gold nanoparticles (indicated by red arrowheads), measuring approximately 15–20 nm, dispersed throughout the scaffold and embedded within the agarose/gelatin matrix.

### 3.3. Scaffolds In Vitro Behavior

The rate at which nanoparticles are released from the scaffold is a crucial factor in assessing their behavior once implanted. Release studies indicate that the GNPs-PEG are slowly released from the AGH-xAu scaffolds when submerged in a physiological solution. The characteristic maximum at 520 nm, associated with plasmon resonance, is observed in the UV–vis spectra of the supernatant resulting from the scaffold immersion (Figure 7a). As expected, the AGH-4Au scaffolds, which contain the highest amount of gold nanoparticles, release larger quantities of these particles (Figure 7b). Regardless of the specific scaffold, both types exhibit a similar release pattern, with around 80% of GNPs-PEG released in the initial 24 h and complete liberation by 48 h.

In examining the cell response to the prepared scaffolds, it is important to consider that the differentiation process of mouse osteoprogenitor MC3T3-E1 cells into osteoblasts, under certain stimuli, occurs in three stages: (a) cell proliferation, (b) matrix maturation, and (c) matrix mineralization. During the early stage (days 1–9), MC3T3-E1 cells replicate actively but do not express alkaline phosphatase (ALP) or form mineralized nodules. In the second stage (days 9–16), the cells begin to produce ALP. By the third phase (after day 16), the cells demonstrate both ALP activity and the formation of mineralized nodules [48]. Matrix maturation and mineralization are typically enhanced by allowing the cells to reach confluence and by adding specific osteogenic factors. Therefore, the early stage is crucial for the transition of premature osteoblasts into mature osteoblasts. [49,50]. In this context, cell adhesion to 3D hydrogels plays a critical role in enabling subsequent cellular processes [51]. Good cell adhesion is facilitated when the surface topography offers multiple binding points, thereby increasing the interface area between the surface and cells [52,53]. Enhanced cell spreading facilitates osteoblast differentiation in osteoprogenitor cells and promotes matrix deposition during bone remodeling [54]. It has been shown that cell shapes influence whether human mesenchymal stem cells commit to osteogenic or adipogenic lineages [55]. Since cell shapes relate to the expression of cytoskeletal proteins and integrins, proper regulation of the cytoskeleton may be essential for the osteoblast differentiation process [56].

In this study, we performed a morphological analysis of MC3T3-E1 cells, as the in vitro shape of the cells significantly impacts the outcome of osteoblast differentiation. Figure 8a presents representative images of the morphology of MC3T3-E1 preosteoblasts adhered to the surfaces of AGH and AGH-4Au scaffolds after 9 days of culture in cell medium without specific differentiation factors. These conditions serve as a control (AGH, without gold nanoparticles) and reflect the maximum concentration of nanoparticles present in the hydrogel AGH-4Au (2.8 μg/g scaffold). In the images, the cell nucleus is stained blue, while F-actin filaments are labelled in red. The cells appear to be firmly anchored to the surfaces of both scaffolds via filopodia, and there is evidence of intercellular communication. Filopodia are thin, finger-like projections of the cell membrane that extend to explore the cellular environment. They bind to extracellular structures, forming adhesion complexes known as focal adhesions. Quantification of these morphological changes was based on cell shape, as measured by the circularity index, and cell spreading, as determined by the mean cell area. [53]. The results in Figure 8b indicate that the circularity index of preosteoblasts cultured on hydrogels loaded with gold nanoparticles (AGH-1Au and AGH-4Au) significantly decreased (** *p* < 0.05) compared to those cultured on hydrogels without nanoparticles (AGH). Conversely, Figure 8c shows a significant increase (* *p* < 0.05) in the mean cell area of preosteoblasts seeded on AGH-4Au hydrogels, compared to those cultured on AGH hydrogels without gold nanoparticles. There were no significant differences in the mean cell area between preosteoblasts seeded onto AGH hydrogels and those seeded onto hydrogels with a lower concentration of nanoparticles (AGH-1Au, 0.7 μg/g scaffold). The cells on AGH-4Au scaffolds exhibited both a significantly increased mean cell area and a significantly decreased circularity index during the early stage of osteoblast differentiation (9 days). These results suggest adequate cell spreading, which may lead to enhanced proliferation and matrix deposition activity, as will be illustrated in Figure 9.

Once mineralization is complete, osteoblasts produce vast extracellular calcium deposits, indicating successful in vitro bone formation. These deposits can be specifically stained bright orange-red using Alizarin Red S. Detection of functional mineralization is commonly used to characterize osteoblasts in vitro. For this reason, the in vitro mineralization process of osteoblasts has been extensively employed as a model for testing the effects of drug treatments and mechanical loading on bone cell differentiation and bone formation [57,58].

In this study, we evaluated the cellular differentiation of preosteoblasts (MC3T3-E1) seeded directly onto the surface of scaffolds: one without any gold nanoparticles (AGH) and two with different concentrations of gold nanoparticles (AGH-1Au and AGH-4Au). Our focus was on the early and late phases of the differentiation process, as these are critical stages. Cell proliferation, representing the early stage, was assessed after 1, 6, and 8 days of culture in basal medium without differentiation factors. Meanwhile, cellular mineralization was evaluated after 15 days under the same experimental conditions. The results, shown in Figure 9a, indicate that preosteoblasts seeded onto the hydrogels, whether in the presence (AGH-1Au and AGH-4Au) or absence (AGH) of gold nanoparticles, were capable of proliferating over time (from days 1 to 8). These facts demonstrate that the final composition of the hydrogels, in terms of the agarose-gelatin-hydroxyapatite ratios and the concentrations of gold nanoparticles used in this study, does not exhibit cytotoxicity toward immature preosteoblasts, allowing for their proliferation. Notably, while there were no significant differences in the proliferation of preosteoblasts seeded onto scaffolds without gold nanoparticles (AGH) compared to those on hydrogels with the lowest concentration of nanoparticles (AGH-1Au), significant differences were observed in the proliferation of preosteoblasts on hydrogels with the higher concentration of nanoparticles (AGH-4Au) compared to those with the lower concentration (AGH-1Au) at both 6 (# *p* < 0.05) and 8 days (^&^ *p* < 0.05). Further, significant differences were also noted when comparing cellular proliferation on AGH-4Au scaffolds with the control condition (* *p* < 0.05), which consisted of hydrogels without nanoparticles (AGH). Thus, it can be concluded that hydrogels with the highest concentration of gold nanoparticles (AGH-4Au, 2.8 μg/g hydrogel) allowed for greater cellular proliferation of immature preosteoblasts, which is essential for the ongoing cellular differentiation process.

In terms of cellular mineralization in the late phase, the results shown in Figure 9b demonstrate significant differences in calcium deposition by preosteoblasts cultured for 15 days on hydrogels with the highest concentration of NPs (AGH-4Au) compared to results obtained from cells cultured on AGH control hydrogels (* *p* < 0.05) and on hydrogels with the lowest concentration of gold nanoparticles (# *p* < 0.05). This study supports Quarles et al.’s classification, indicating that the early stage of osteoblast differentiation is a critical transition from immature to mature osteoblasts. Various studies have indicated that gold nanoparticles ranging from 5 to 20 nm in size enhance osteoblastic proliferation and differentiation, as these nanoparticles interact with the plasma membrane and influence the metabolic MAPK (p38 mitogen-activated protein kinase) pathway regulating osteogenic genes [35,59,60,61]. The findings regarding cellular responses to the materials underscore the significance of rapid gold NPs release (24–48 h) from the scaffolds (Figure 7), which promotes the early phase of differentiation from immature preosteoblasts to mature osteoblasts, which is a vital stage that is placed between days 1 and 9.

This study establishes a baseline for incorporating gold nanoparticles into this system and identifies a concentration that yields positive and significant outcomes. Nanoparticles embedded in these hydrogels may serve as an excellent method to enhance the functionality of the composite scaffold, demonstrating therapeutic potential for bone tissue engineering by promoting the viability, proliferation, and mineralization of osteoprogenitor MC3T3 cells.

## 4. Conclusions

A three-dimensional bone tissue-engineered scaffold with osteogenic properties using only mild aqueous conditions, without any toxic chemicals, was fabricated through the GELPOR3D method. This technique enables the direct incorporation of labile molecules during scaffold fabrication at low temperatures. In this work, we demonstrated that spherical gold nanoparticles (14 nm) can be homogeneously dispersed throughout the scaffolds. To enhance the stability of these gold nanoparticles in suspension and facilitate their inclusion in the scaffolds, we functionalized them with thiolated polyethylene glycol.

The nanoparticles are gradually released into a physiological medium, with complete release occurring after 48 h. Our findings highlight the significant potential of gold nanoparticles as bioactive agents for bone regeneration. Scaffolds containing these nanoparticles exhibited a marked improvement in adhesion, proliferation, and mineralization of preosteoblasts (MC3T3 cells) at the concentrations studied. The results confirm that the early stages of osteoblast differentiation are critical for the transformation from premature to mature osteoblasts. Future studies are needed to evaluate different concentrations of gold nanoparticles in order to determine the optimal dosage that provides significant benefits without compromising cell viability.

## Figures and Tables

**Figure 1 pharmaceutics-17-01103-f001:**
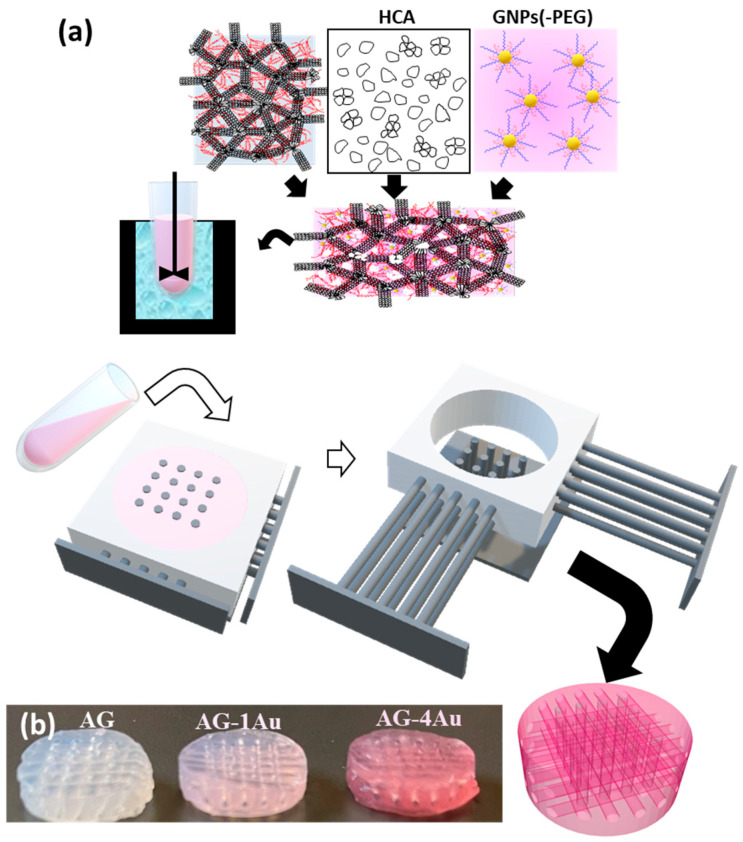
(**a**) Scaffold fabrication method: GELPOR3D; (**b**) HCA-free scaffolds with increasing gold nanoparticle content.

**Figure 2 pharmaceutics-17-01103-f002:**
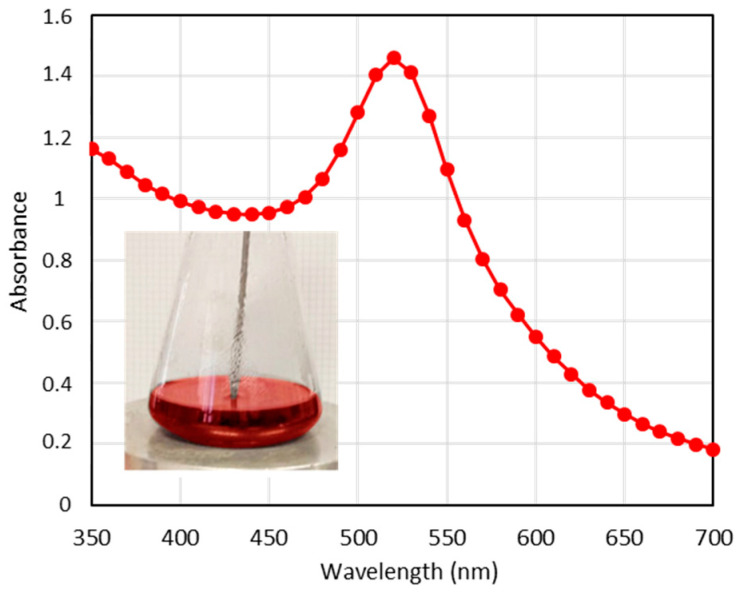
UV–vis spectra corresponding to the GNPs. (inset): Photograph of GNPs suspension during the synthesis process.

**Figure 3 pharmaceutics-17-01103-f003:**
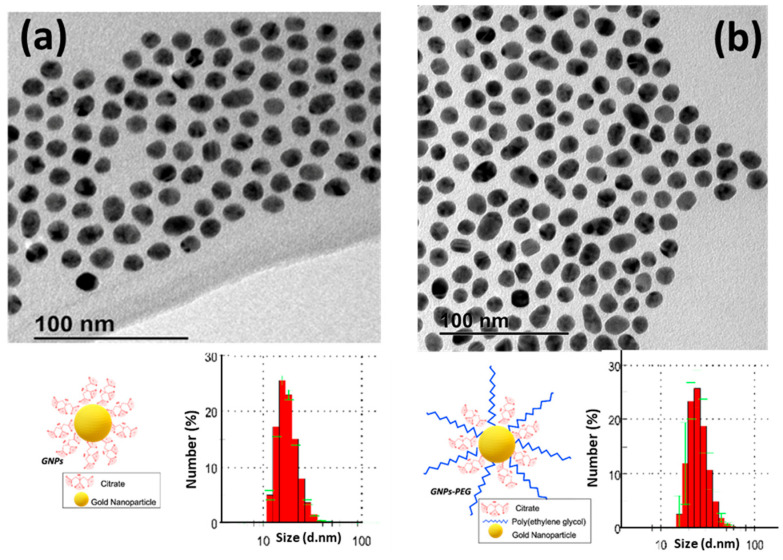
TEM images and size histograms measured by DLS (inset) of the two types of gold nanoparticles synthesized: (**a**) GNPs; (**b**) GNPs-PEG. The green lines in the histograms represents the error bars.

**Figure 4 pharmaceutics-17-01103-f004:**
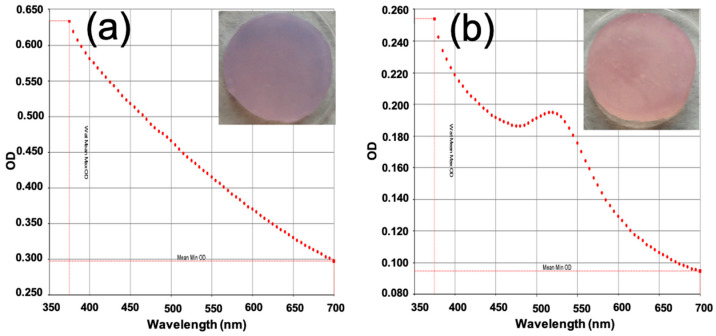
UV–vis absorption spectra of the supernatant solution remaining after 12 h of immersion in PBS of HCA-free scaffolds containing (**a**) GNPs or (**b**) GNPs-PEG.

**Figure 5 pharmaceutics-17-01103-f005:**
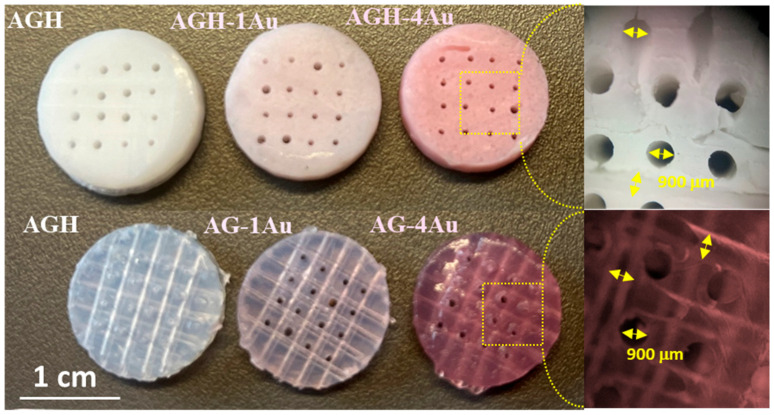
Photographs of designed porous scaffolds fabricated with varying GNPs-PEG content: without (**bottom image**) and with ceramic (**top image**).

**Figure 6 pharmaceutics-17-01103-f006:**
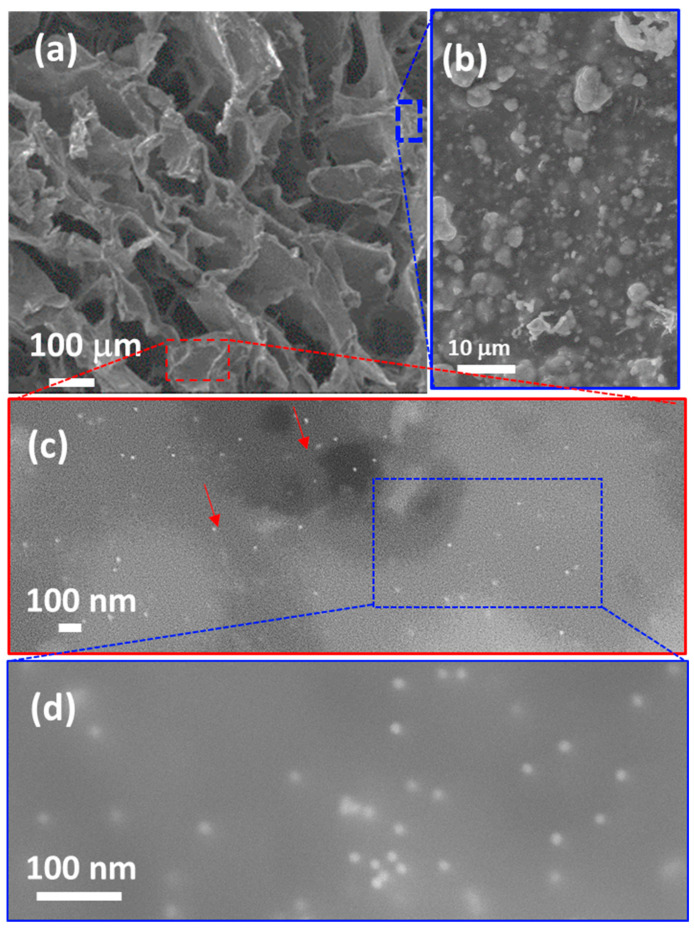
(**a**,**b**) SEM images and (**c**,**d**) backscattered electron SEM images of the AGH-4Au scaffold cross-section. Red arrows point gold nanoparticles.

**Figure 7 pharmaceutics-17-01103-f007:**
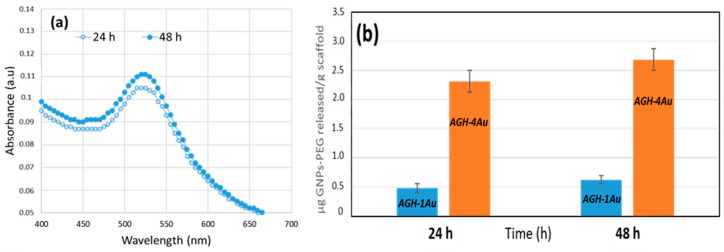
Release of GNPs-PEG from the scaffolds in PBS at 37 °C as a function of time. (**a**) UV–vis spectra of the supernatant at various time points for the AGH-1Au scaffold immersion; (**b**) cumulative amount released as a function of time from different scaffolds.

**Figure 8 pharmaceutics-17-01103-f008:**
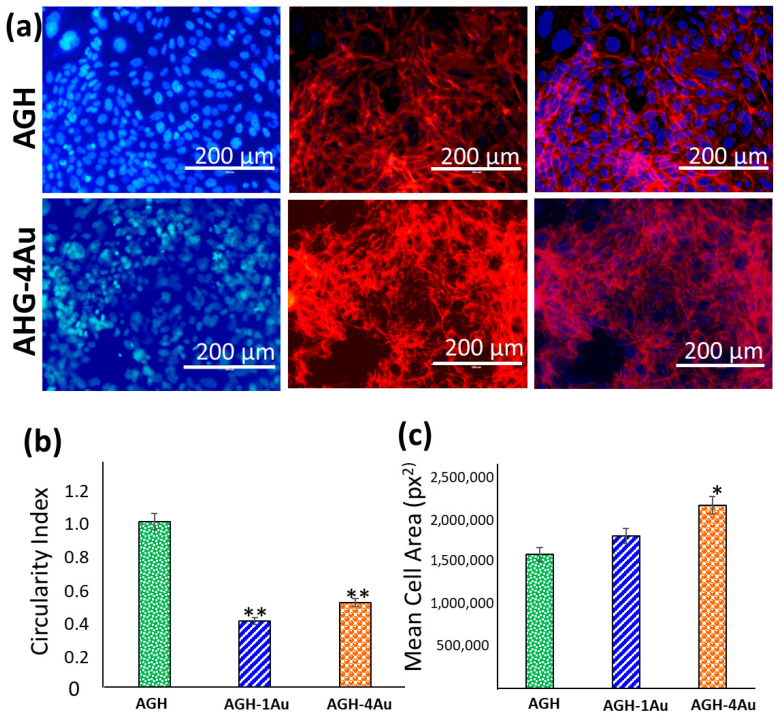
Cell adhesion and morphology of MC3T3-E1 preosteoblasts seeded onto AGH (without NPs) and AGH-4Au hydrogels after 9 days of culture. (**a**) Representative images obtained by fluorescence laser scanning microscopy. Atto 565–phalloidin and DAPI were used as fluorescence probes to stain the F-actin microfilaments and cell nucleus, respectively. Quantification of cell morphology in terms of (**b**) circularity index and (**c**) mean cell area (pixel^2^) of MC3T3-E1 preosteoblasts cultured onto AGH, AGH-1Au, and AGH-4Au hydrogels. Error bars are standard error of the mean, where n = 50. * *p* < 0.05 and ** *p* < 0.01, statistical differences *vs* AGH condition.

**Figure 9 pharmaceutics-17-01103-f009:**
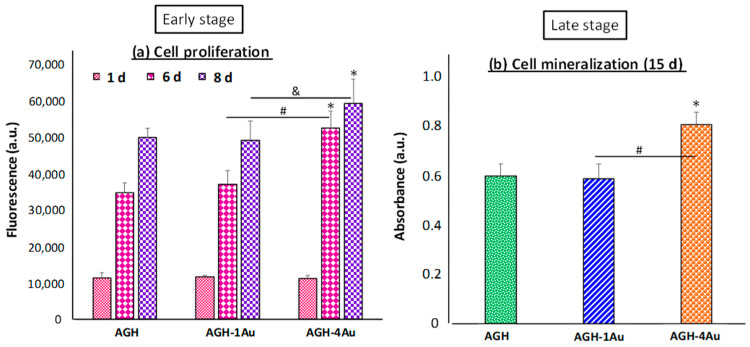
Cell differentiation process of MC3T3-E1 preosteoblasts seeded onto AGH (without NPs) and AGH-1Au and AGH-4Au hydrogels. (**a**) Cell proliferation (early stage) of preosteoblasts cultured for 1, 6, and 8 days measured by alamar blue test. (**b**) Cell mineralization (late stage) of preosteoblasts after 15 days of culture evaluated by alizarin red staining. * *p* < 0.05, statistical differences vs. AGH condition. ^#^ *p* < 0.05, AGH-1Au vs. AGH-4Au at 6 days in cell proliferation and at 15 days in cell mineralization. ^&^ *p* < 0.05 AGH-4Au vs. AGH-1Au after 8 days of proliferation.

## Data Availability

All data supporting the findings of this study are included in the present manuscript. Further inquiries can be directed to the corresponding author.

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
