# Peer review of "Osteogenic-Inducing Apatite/Agarose/Gelatin Hybrid Scaffolds Embedding Gold Nanoparticles"

_pharmaceutics, 2025, doi:10.3390/pharmaceutics17091103_

Round 1

Reviewer 1 Report

Comments and Suggestions for Authors

1) In line 34, the authors correctly point out that an ideal scaffold for bone tissue engineering should also be biodegradable. However, this important aspect hasn't been addressed in the study. It would be worthwhile to include some degradation tests under physiological conditions to better assess the potential biodegradability of this scaffold.

2) The authors mention that adding hydroxyapatite improves the mechanical properties of the scaffold, but this claim hasn’t been supported by any testing. To justify the inclusion of the ceramic component, it would be important to include some mechanical evaluations, such as compression tests.

3) It is unclear whether the samples for SEM morphological analysis were cross-sectioned using cryo-fracture after immersion in liquid nitrogen, or if the analysis only refers to the surface morphology.

4) It would be helpful if the authors specified the sample preparation method for GNPs used in TEM analysis, as this information is currently missing.

5) In line 204, the authors report the average size of the nanoparticles along with the error. It would be helpful to also indicate how many particles were measured (N = number of nanoparticles).

6) In Figure 6a and 6b, it is not clear what magnification is represented by the blue lines. It might be helpful to use two distinct colours to clearly identify the two different areas.

7) Figure 7b shows the release of GNPs-PEG after 24 and 48 hours of incubation under physiological conditions. To better visualize the data, it would be more effective to report the percentage of release relative to the amount of NPs loaded into the scaffold vs time, rather than the absolute amount of GNPs-PEG released in micrograms.

8) In line 345, the authors mention that they quantified morphological changes based on cell shape using the circularity index, but they don’t explain which method or software they used for this analysis. Also, in the legend of Figure 8, it says the data come from just 10 cells (N=10). To get reliable results from morphological analysis, they should analyze a larger number of cells.

9) In line 182, the authors state that fluorescence was measured to determine mineralization, while in Figure 9b an absorbance measurement is shown. This discrepancy should be clarified

10) At line 145, the symbol λ is missing. At line 167, the incubation temperature in the oven is not specified. 

Comments on the Quality of English Language

Please review the English, as some sentences are unclear.

Author Response

Open Review

(x) I would not like to sign my review report

( ) I would like to sign my review report

Quality of English Language

(x) The English could be improved to more clearly express the research.

( ) The English is fine and does not require any improvement.

Yes      Can be improved        Must be improved      Not applicable

The manuscript has been reviewed to improve the quality of the English language

Does the introduction provide sufficient background and include all relevant references?

(x)       ( )         ( )        ( )

Is the research design appropriate?

( )        (x)        ( )        ( )

Are the methods adequately described?

( )        (x)        ( )        ( )

Are the results clearly presented?

( )        (x)        ( )        ( )

Are the conclusions supported by the results?

(x)       ( )         ( )        ( )

Are all figures and tables clear and well-presented?

( )        (x)        ( )        ( )

Comments and Suggestions for Authors

1) In line 34, the authors correctly point out that an ideal scaffold for bone tissue engineering should also be biodegradable. However, this important aspect hasn't been addressed in the study. It would be worthwhile to include some degradation tests under physiological conditions to better assess the potential biodegradability of this scaffold.

Our objective in this work was to homogeneously integrate GNPs within the gelatin/agarose/apatite scaffolds whose surgical handling, mechanical performance, integration and progressive degradation have been demonstrated when implanted subcutaneously

  1. García-Honduvilla, A. Coca, M. A. Ortega, C.Trejo, J. Román, J. Peña, M.V.Cabañas, J.Buján and M. Vallet-Regí (2018). Improved Connective Integration of a Degradable 3D-Nano-apatite/Agarose Scaffold Subcutaneously Implanted in a Rat Model. Journal of Biomaterials Applications. 33(5),741–752.

In order to clarify this item and include the above-mentioned reference (we missed it), we have included (line 61) the following sentence at the end of the paragraph:

“These scaffolds demonstrate excellent surgical handling, mechanical performance, integration, and gradual degradation when implanted subcutaneously in rats [21], showing no premature decomposition that could compromise their performance [12, 22]”.

2) The authors mention that adding hydroxyapatite improves the mechanical properties of the scaffold, but this claim hasn't been supported by any testing. To justify the inclusion of the ceramic component, it would be important to include some mechanical evaluations, such as compression tests.

In that sentence, we aimed to convey that the incorporation of a ceramic within hydrogel-based scaffolds significantly enhances one of their primary drawbacks: low mechanical performance. In this sense, we have modified the sentence in line 49:

“Adding calcium phosphates, such as hydroxyapatite, further improves the mechanical performance and bioactivity of hydrogel-based scaffolds, promoting osteoinduction and osteoconduction”.

We want to emphasise the use of mechanical "performance" instead of properties since we have not carried out a complete and meticulous mechanical characterisation. Despite this, our scaffolds can be easily handled by the surgeon, adapted to the defect if necessary, and implanted, thus demonstrating adequate mechanical performance, as shown in the previously mentioned reference.

3) It is unclear whether the samples for SEM morphological analysis were cross-sectioned using cryo-fracture after immersion in liquid nitrogen, or if the analysis only refers to the surface morphology.

SEM images in Figure 6 correspond to the internal morphology of the scaffold. To observe this, the hydrated samples were cross-sectioned using a cutter, lyophilised, and prepared for SEM observation. To clarify this point, the caption for Figure 6 has been modified.

“Figure 6. (a, b) SEM images and (c, d) backscattered electron SEM images of the AGH-4Au scaffold cross-section.”

4) It would be helpful if the authors specified the sample preparation method for GNPs used in TEM analysis, as this information is currently missing.

We have included this information in the manuscript, line 217:

“To determine the size and shape of the GNPs by TEM a drop of an aqueous suspension containing the nanoparticles was placed on a copper grid and allowed to evaporate.”

5) In line 204, the authors report the average size of the nanoparticles along with the error. It would be helpful to also indicate how many particles were measured (N = number of nanoparticles).

130 nanoparticles were measured; this data has been included in the manuscript, line 221.

6) In Figure 6a and 6b, it is not clear what magnification is represented by the blue lines. It might be helpful to use two distinct colours to clearly identify the two different areas.

We have modified the lines that indicate the progressive magnification shown in this figure; we hope that it appears more clearly now.

7) Figure 7b shows the release of GNPs-PEG after 24 and 48 hours of incubation under physiological conditions. To better visualize the data, it would be more effective to report the percentage of release relative to the amount of NPs loaded into the scaffold vs time, rather than the absolute amount of GNPs-PEG released in micrograms.

We agree with the referee regarding the relevance of the relative amount of NPs loaded, and for this reason, we mention these values in the text. However, despite the considerable amount released by the highly loaded samples (AGH-4Au) no deleterious effects have been observed in the in vitro studies. The use of relative percentages in this figure does not visualise the differences between the two samples. In addition, the paragraph has been reorganized as follows:

“The rate at which nanoparticles are released from the scaffold is a crucial factor in assessing their behaviour once implanted. Release studies indicate that the GNPs-PEG are slowly released from the AGH-xAu scaffolds when submerged in a physiological solution. The characteristic maximum at 520 nm, associated with plasmon resonance, is observed in the UV-Vis spectra of the supernatant resulting from the scaffold immersion (Figure 7a). As expected, the AGH-4Au scaffolds, which contain the highest amount of gold nanoparticles, release larger quantities of these particles (Figure 7b). Regardless of the specific scaffold, both types exhibit a similar release pattern, with around 80% of GNPs-PEG released in the initial 24 hours and complete liberation by 48 hours.”

8) In line 345, the authors mention that they quantified morphological changes based on cell shape using the circularity index, but they don't explain which method or software they used for this analysis. Also, in the legend of Figure 8, it says the data come from just 10 cells (N=10). To get reliable results from morphological analysis, they should analyze a larger number of cells.

The authors appreciate the reviewer's comment, and in order to clarify this information, a new paragraph has been included in the Materials and Methods section (2.3.2.2. Cell Morphology and adhesion assay), as follows:

“For fluorescence microscopy, an EVOS FL Cell Imaging System inverted optical microscope equipped with three types of LED light (IEX (nm); IEM (nm)) from AMG (Advanced Microscopy Group, Bothell, WA, USA) was used. The red channel was used to observe the cytoskeleton labelled with the probe Phalloidin-rhodamine conjugate (540/565), and the blue channel to observe the cell nucleus labelled with DAPI (358; 461).

For quantitative analysis of cell morphology, images were captured from triplicate samples (10 randomly chosen fields of view each) on a given number (n = 50) of randomly chosen cells. Cell circularity is used to provide a quantification of cell shape. The formula used for determining the circularity index (CI) is CI = 4π*(Area/Perimeter^2). A value of 1.0 indicates a perfect circle and a value of 0.0 a totally elongated structure.”

Also, the legend of Figure 8 has been also corrected: Error bars are standard error of the mean, where n = 50.

9) In line 182, the authors state that fluorescence was measured to determine mineralisation, while in Figure 9b, an absorbance measurement is shown. This discrepancy should be clarified

The authors apologise for the error in line 182 of the Materials and Methods section (2.3.2.4. Mineralisation assay), and the sentence, line 196, has been corrected as follows:

 “and the absorbance was measured at 620 nm using the SynergyTM 4 reader (Biotek).”

10) At line 145, the symbol λ is missing. At line 167, the incubation temperature in the oven is not specified.

We have included both items.

Reviewer 2 Report

Comments and Suggestions for Authors

The manuscript is dedicated to the synthesis and study of the composite gel on the base of agarose with the inclusion of ceramic and gold particles as a promising implant for bone tissue regeneration. The investigation is well performed, the choice of methods is classical for this type of study. The researchers show that the offered composition works well for the enhancement of osteoblasts growth. A few questions and comments:

  1. It is mentioned that the gel is formed by the temperature influence. The agarose is heated to 90 degrees C, then cooled to 40 degrees C to incorporate other components, then cooled to room\normal body temperature. Does it mean that the scaffold if kind of liquid at 40 degrees C? It is temperature that can be reached in human  body during inflammation. Please comment on this point.
  2. Did you study the biodegradation of the composite? Would it decompose after the implantation?
  3. Do the gold NPs need to be released in the outer solution? Could it be more effective to immobilize it (by conjugation, for example) on the gel matrix chains?
Comments on the Quality of English Language

Quality of English language should be substantially increased, especially in the abstract. There are several typos, wrong use of verb tense, same words in one sentence, etc. Please read the manuscript carefully.

Author Response

Open Review

(x) I would not like to sign my review report

( ) I would like to sign my review report

Quality of English Language

(x) The English could be improved to more clearly express the research.

( ) The English is fine and does not require any improvement.

Yes      Can be improved        Must be improved      Not applicable

Does the introduction provide sufficient background and include all relevant references?

(x)       ( )         ( )        ( )

Is the research design appropriate?

(x)       ( )         ( )        ( )

Are the methods adequately described?

(x)       ( )         ( )        ( )

Are the results clearly presented?

(x)       ( )         ( )        ( )

Are the conclusions supported by the results?

(x)       ( )         ( )        ( )

Are all figures and tables clear and well-presented?

(x)       ( )         ( )        ( )

Comments and Suggestions for Authors

The manuscript is dedicated to the synthesis and study of the composite gel on the base of agarose with the inclusion of ceramic and gold particles as a promising implant for bone tissue regeneration. The investigation is well performed, the choice of methods is classical for this type of study. The researchers show that the offered composition works well for the enhancement of osteoblasts growth. A few questions and comments:

It is mentioned that the gel is formed by the temperature influence. The agarose is heated to 90 degrees C, then cooled to 40 degrees C to incorporate other components, then cooled to room\normal body temperature. Does it mean that the scaffold if kind of liquid at 40 degrees C? It is temperature that can be reached in human body during inflammation. Please comment on this point.

Regarding the state of the scaffold at 40 °C, we would like to clarify that once it has solidified, it requires a higher temperature (>60 °C) to begin softening and turning into a viscous suspension. This is due to the hysteresis that characterises the thermogelling behaviour of agarose; in addition, the presence of the ceramic component makes it difficult to return to the liquid or, at least, a paste state. In fact, the release experiments are carried out at 37 °C with no evidence of phase change.

Anyway, we would like to emphasize that the scaffold preparation is carried out at room temperature; in this sense the manuscript, line 119, has been slightly modified to clarify this point:  

“Once the slurry was homogeneous, it was poured into a mould and allowed to gel at room temperature for approximately 5 minutes. Subsequently, the filaments were withdrawn and the resulting 3D interconnected porous scaffold removed [43]. (Figure 1).”

Did you study the biodegradation of the composite? Would it decompose after the implantation?

Not in this paper, but we have previously published a few papers that demonstrate that their implantation subcutaneously

  1. García-Honduvilla et al. (2018). Improved Connective Integration of a Degradable 3D-Nano-apatite/Agarose Scaffold Subcutaneously Implanted in a Rat Model. Journal of Biomaterials Applications. 33(5),741–752),

in rabbit radius bone defect

García-Lamas, L. et al. In Vivo Behavior in Rabbit Radius Bone Defect of Scaffolds Based on Nanocarbonate Hydroxyapatite. J. Biomed. Mater. Res. - Part B Appl. Biomater. 2024, 112, 1–14

or in chicken embryos grown ex ovo

Paris, J.L.; Lafuente-Gómez, N.; Cabañas, M.V.; Román, J.; Peña, J.; Vallet-Regí, M. Fabrication of a Nanoparticle-Containing 3D Porous Bone Scaffold with Proangiogenic and Antibacterial Properties. Acta Biomater. 2019, 86, 441–449

show a progressive scaffold degradation with no evidence of decomposition that may compromise the scaffold's performance.

In this sense, we have included (line 61) the following sentence at the end of the paragraph.

“These scaffolds demonstrate excellent surgical handling, mechanical performance, integration, and gradual degradation when implanted subcutaneously in rats [21], showing no premature decomposition that could compromise their performance [12, 22].”

Do the gold NPs need to be released in the outer solution? Could it be more effective to immobilize it (by conjugation, for example) on the gel matrix chains?

We considered the bibliography about the use of gold nanoparticles to stimulate the osteogenic differentiation of osteoprogenitor cells for improving bone tissue engineering  

Changqing et al.  Gold Nanoparticles Promote Ostogenic Differentiation of Mesenchymal StemCells through P38 MAPK Pathway. ACS Nano 2010, 4, 6439–6448; Li et al. Advances in the Application of Gold Nanoparticles in Bone Tissue Engineering. J. Biol. Eng. 2020, 14, 1–15, doi:10.1186/s13036-020-00236-3; Zhang et al. PEGylated Gold Nanoparticles Promote Osteogenic Differentiation in in Vitro and in Vivo Systems. Mater. Des. 2021, 197, 109231, doi:10.1016/j.matdes.2020.109231).

Briefly, both strategies: NPs immobilisation vs. release to the cell medium have demonstrated the ability to regenerate bone

Kearney et al. Switchable Release of Entrapped Nanoparticles from Alginate Hydrogels. Adv Healthc Mater. 2015;4(11):1634-1639.  doi: 10.1002/adhm.201500254; Xia et al. Gold nanoparticles in injectable calcium phosphate cement enhance osteogenic differentiation of human dental pulp stem cells. Nanomedicine: Nanotechnology, Biology and Medicine,14, 1,2018, 35-45,https://doi.org/10.1016/j.nano.2017.08.014; Abdelrasoul et al. Nanocomposite scaffold fabrication by incorporating gold nanoparticles into biodegradable polymer matrix: Synthesis, characterization, and photothermal effect,Materials Science and Engineering: C, 56,2015, 305-310,https://doi.org/10.1016/j.msec.2015.06.037; Navaei et al. Gold nanorod-incorporated gelatin-based conductive hydrogels for engineering cardiac tissue constructs, Acta Biomaterialia,Volume 41,2016,Pages 133-146, https://doi.org/10.1016/j.actbio.2016.05.027; Kim et al. A 3D  calcium-deficient hydroxyapatite-based scaffold with gold nanoparticles effective against Micrococcus luteus as an artificial bone substitute Materials & Design, Volume 219, 2022,110793, https://doi.org/10.1016/j.matdes.2022.110793. 

However, since these experiments were conducted under different conditions, it is challenging to compare them and establish the most effective strategy. Considering that our scaffolds are based on a highly hydrated ceramic-reinforced hydrogel, we decided to try the NPs release approximation. In our work, the results obtained, in terms of the cellular response to the scaffolds demonstrate the importance of the quick gold NPs release (24-48 h) from the scaffolds, promoting the early phase of the cell differentiation process from immature pre-osteoblasts to mature osteoblasts which is a vital stage that takes place between days 1-9.

Comments on the Quality of English Language

Quality of English language should be substantially increased, especially in the abstract. There are several typos, wrong use of verb tense, same words in one sentence, etc. Please read the manuscript carefully.

The manuscript has been reviewed to improve the quality of the English language

Submission Date

21 July 2025

Date of this review

06 Aug 2025 12:14:15

Round 2

Reviewer 1 Report

Comments and Suggestions for Authors

None